# Metabolomics analysis of the yolk of Zhijin white goose during the embryogenesis based on LC-MS/MS

Zhonglong Zhao[1,2], Hong Yang[3], Zhiwei Wang[1,2], Zhaobi Ai[1,2], Runqian Yang[1,2], Zhong Wang[1,2], Tiansong Wang[4], Kaibin Fu[1,2], Yong Zhang[1,2]*

1 Key Laboratory of Animal Genetics, Breeding and Reproduction in the Plateau Mountainous Region, Ministry of Education, College of Animal Science, Guizhou University, Guiyang, Guizhou, People's Republic of China, 2 Guizhou Provincial Key Laboratory of Animal Genetics, Breeding and Reproduction, College of Animal Science, Guizhou University, Guiyang, People's Republic of China, 3 Bijie City Animal Husbandry Station, Bijie, Guizhou, People's Republic of China, 4 Agricultural College, Tongren Polytechnic College, Tongren, Guizhou, People's Republic of China

* GZDXZY2020@163.com

**Data Availability Statement:** All relevant data are within the paper and its Supporting information files.

## Abstract

The egg yolk of the goose is rich in lipids, proteins and minerals, which is the main source of nutrition during the goose embryogenesis. Actually, the magnitude and variety of nutrients in yolk are dynamically changed to satisfy the nutritional requirements of different growth and development periods. The yolk sac membrane (YSM) plays a role in metabolizing and absorbing nutrients from the yolk, which are then consumed by the embryo or extra-fetal tissues. Therefore, identification of metabolites in egg yolk can help to reveal nutrient requirement in goose embryo. In this research, to explore the metabolite changes in egg yolk at embryonic day (E) 7, E12, E18, E23, and E28, we performed the assay using ultra-high performance liquid chromatography/tandem mass spectrometry (UHPLC-MS/MS). The findings showed that E7 and E12, E23 and E28 were grouped together, while E18 was significantly separated from other groups, indicating the changes of egg yolk development and metabolism. In total, 1472 metabolites were identified in the egg yolk of Zhijin white goose, and 636 differential metabolites (DMs) were screened, among which 264 were upregulated and 372 were downregulated. The Kyoto Encyclopedia of Genes and Genomes (KEGG) pathway analysis showed that the DMs were enriched in the biosynthesis and metabolism of amino acids, digestion and absorption of protein, citrate cycle (TCA cycle), aminoacyl-tRNA biosynthesis, phosphotransferase system (PTS), mineral absorption, cholesterol metabolism and pyrimidine metabolism. Our study may provide new ideas for improving prehatch embryonic health and nutrition.

## Introduction

Geese were domesticated by humans about 7,000 years ago and gradually became one of the most important poultry [1]. Zhijin white goose is a well-known local breed in Guizhou Province that is famous for white feathers, strong tolerance to rough feeding and stress resistance.

**Funding:** The Science and Technology Project of Guizhou Province, China (No. QKHZC[2021]YB156 and No. QKHZC[2020]1Y047).

**Competing interests:** The authors have declared that no competing interests exist.

In the past decades, the production performance of geese were dramatically increased by modern formulas and advanced technology, which indicated a significant advancement in goose industry. Nevertheless, only a few articles have studied the significance of embryo nutrition in goose, which is well acknowledged to affect the health, hatching quality and hatch capacity of the goose embryos.

During goose embryogenesis, nutrients required for embryonic growth and development are originated from egg yolk and albumin [2], which meet nutritional requirements basically. Egg yolk, which mainly derived from the deposition of lipids and proteins from the liver to the ovary, is the primary nutrient supplier for embryonic development [3]. The yolk lipids are the main provider of energy, more than 90% of the energy needed by embryos and hatchlings is obtained from the oxidation of egg lipids during the late hatching and early hatchling stages [4]. About 66% of the egg yolk dry matter is made up of lipids, including 62% triglycerides, 33% phospholipids and less than 5% cholesterol [5]. Polyunsaturated fatty acids (especially Eicosapentaenoic Acid, EPA and Docosahexaenoic Acid, DHA) and phosphatidylcholine are essential for nerve and brain development and are greatly enriched in the yolk during early embryogenesis [6–8]. Yolk protein is one of the crucial components of yolk, accounting for about 16%-17% of the total weight of yolk, The proteins in yolk are mainly in the form of lipoproteins, such as low-density lipoproteins (LDLs) and high-density lipoproteins (HDLs). Phosvitin and IgY, two representative egg yolk proteins, are recognized because they have a variety of biological functions, including antibacterial, anticancer, antioxidant and immunomodulatory properties [9, 10]. Compared to egg white, the yolk contains more nutrients such as minerals and fat-soluble vitamins, etc., which provides a foundation for the normal development of goose embryos.

Actually, due to the continuous metabolism of the yolk sac membrane (YSM), the nutrients in the yolk undergo dynamic changes at different periods of embryogenesis. The YSM, which forms from the hindgut of the embryo, has a highly vascularized structure [11]. The ability of the embryo to utilize yolk nutrients is affected by the ability of the YSM to transport nutrients [12]. The nutrients in the yolk are transferred from the yolk sac to the embryonic circulation by the YSM which is not completely developed during the first week of incubation. At this point, the main nutrient that is digested and absorbed is glucose [13]. The nutrients in yolk were metabolized by YSM in two indirect method: 1) the yolk sac synthesizes and secretes digestive enzymes for release into yolk; and 2) the yolk sac can also absorb nutrients for intracellular metabolism and carry metabolites to the yolk [14–16]. Therefore, studying the yolk metabolites at different periods of goose embryo development is of great significance to understand the nutritional requirements and developmental rules of goose embryos. From fertilized eggs, we collected six yolk samples at each stage (E7, E12, E18, E23 and E28) and conducted LC-MS/MS analysis to investigate the metabolic variation of yolk nutrients during embryonic growth and development.

## Materials and methods

The Institutional Animal Care and Use Committee of Guizhou University, Guiyang, China approved all procedures and animal care (approval number: EAE-GZU-2023-E008). The trial was conducted in strict accordance with relevant guidelines and regulations. The study was carried out under European Union animal protection standards. Every effort was made to ensure that the animals suffered as little as possible.

### Animals and sample collection

All the Zhijin white goose eggs used in the trail originated from the same farms (Dafang County Jingdong Breeding Co., Bijie, China) (105.545403E, 27.090199N). A total of 150 eggs

of Zhijin white goose with uniform appearance were purchased. The Zhijin white geese were 35 weeks old and the feeding standards were consistent. 150 goose eggs were incubated at variable temperatures. On the sixth day of incubation, all the eggs were illuminated to identify the fertilization status and 31 unfertilized eggs were excluded. Six samples of E7, E12, E18, E23 and E28 were collected on each sampling day, respectively. On the sampling day, we gently opened the goose eggs from the air cell with clean surgical scissors, and then slowly peeled off the eggshell membrane with sterilized tweezers to reveal the embryo and yolk sac membrane. A ten-milliliter syringe was used to pierce the yolk sac membrane and suck out two-milliliter yolk, which was then completely mixed before freezing with liquid nitrogen for further untargeted metabolomic analysis.

## Metabolites extraction form egg yolk

Briefly, the samples were gradually melted at 4°C. A prechilled acetonitrile/methanol/ aqueous solution (2:2:1, v/v) is added to appropriate volumes of samples, followed by vortex and ultrasonication for 30min at 5°C. Subsequently, the extraction was allowed to settle at -20°C for 10 min. Next, the samples were centrifuged at 14000g for 20 min. The supernatant was collected and placed in an EP tube and dried by a nitrogen flow. Mass spectrometry analysis with addition of 100μL acetonitrile solution was used for vortex and redissolution, centrifuged at 14000g, 4°C for 15 min. The supernatant was eventually added to the system for subsequent testing [17]. For the regulation and quality control (QC) of the system, QC samples were obtained and mixed thoroughly by taking all samples of equal volumes, and analyzed using the same method.

## UHPLC-Q-exactive orbitrap MS

An UHPLC system (Vanquish UHPLC, Thermo) was used in conjunction with an Orbitrap Q Exactive mass spectrometer (Thermo Fisher, Germany) for the UHPLC-MS/MS analysis in Shanghai Personal Bio Technology Co., Ltd. (Shanghai, China). The column temperature was adjusted to 25°C. The added volume of tested samples was 2 μL and the flow rate was regulated to 0.3 mL/min. The water, 25 mM ammonium acetate and 25 mM ammonia were mixed to form mobile phase A and acetonitrile was used as mobile phase B. The gradient of solvent was adjusted in the following way: from 0 to 1.5 min, 98% B; from 1.5 to 12min, 98% to 2% B; from 12 to 14 min, 2% B; from 14 to 14.1min, 2% to 98% B; from 14.1 to 17 min, 98% B to balance the systems.

In the process of throughout analysis, the samples were added in a 4 °C automatic injector. To reduce the interference of equipment signal fluctuations, the random sequences were selected for consecutive analyses of the samples, while QC samples were processed in the same way. After separation, the extracts were processed by the Q-Exactive Orbitrap MS combined with electrospray ion source working in positive (POS) and negative (NEG) ionization mode. The preferred parameters were set as shown below: auxiliary gas 1, 60psi; auxiliary gas 2, 60psi; curtain gas, 30psi; The ion source temperature was set at 600°C, and the ion spray floating voltage at NEG and POS was −5500V and +5500V, respectively.

## Data processing

After the UHPLC-MS/MS analysis, the raw data was transformed into mzXML files using ProteoWizard MSConvert and peak alignment, retention time adjustment and peak area detection were performed using XCMS software [18]. The following parameters were used for peak picking: centWave m/z = 10 ppm, peak width = c (10,60), prefilter = c (10,100). The main parameters were set as shown: bw = 5, mzwid = 0.025, minfrac = 0.5 were used for peak grouping. The

isotopes and adducts were annotated with CAMERA (Collection of Algorithms of Metabolite pRofile Annotation). In the features of the extracted ions, only at least one set of variables with nonzero measurements greater than 50% are retained. The accuracy of the m/z values (<10ppm) was used to identify the compounds and MS/MS spectra with an internally established database with available true criteria for metabolites.

### Statistical analysis

The R packages (ropls) were used to analyze the normalized data. Multivariate data analysis was performed, including principal component analysis (PCA) and sparse partial least-squares discriminant analysis (sPLS-DA). The stability of the model was assessed using 7-fold cross-validation and response permutation testing. The projected importance (VIP) value of each variable in the OPLS-DA model is calculated to represent its contribution to the classification. To determine the significance of the difference between the two groups of independent samples, the student's t-test was used. Metabolites that were statistically meaningful between groups were screened according to the variable importance in projection (VIP) value>1 and $P$ value<0.05 from the OPLS-DA model. Based on the Kyoto Encyclopedia of Genes and Genomes (KEGG) database (http://www.genome.jp/kegg/), which mapped the function and metabolic pathways of differential metabolites. Statistical analysis was performed using MetaboAnalyst 5.0 (https://www.metaboanalyst.ca/).

## Results

### Method validation and multivariate statistical analysis of LC-MS

The various metabolites in the YS were identified by UHPLC-Q-Exactive Orbitrap mass spectrometry combined with broad-scale ion scanning spectrometry. The total ion chromatogram (TIC) of six QC samples were compared a by spectral overlap (Fig 1A and 1B). The results indicated that the response intensity and residence time of each spectrum peak generally coincided. After the metabolites characteristics with relative standard deviation (RSD) of QC >30% were dropped and residual proportion of peak >70% (Fig 1C and 1D). This suggested that the reliability and repeating accuracy of the LC-MS system method was adequate. To assess the dimensional reduction performance of the assay platform and to visualize the extracted metabolites, the unsupervised PCA was carried out with the total metabolites during the five hatching stages. As shown in Fig 2A and 2B, all sample spots were displayed within the 95% confidence interval ellipse by the first principal component and the second principal component under the positive and negative modes. The six QC samples were pooled in the central area of the score plot, implying the separation of different groups has more to do with biological variation than with analytical variation. Under both modes, E18 was significantly separated from the other groups, while E7 was close to E12 and E23 was close to E28.

The supervised sPLS-DA was used for further detailed analysis of the variables contributing to sample classification, and a significant trend towards separation was observed between all groups (Fig 2C and 2D). Overall, the repeatability and reliability of the data generated in this trial were reasonable for further studies. Therefore, a total of 17 superclass (S1 Table) 1472 metabolites were detected in the egg yolk of Zhijin white goose, which contained 840 metabolites and 632 metabolites in the POS and NEG modes, respectively (S2 Table). Concretely, the superclasses metabolites that account for the majority are lipids and lipid-like molecules, Organic acids and derivatives and Organoheterocyclic compounds in order (Fig 3).

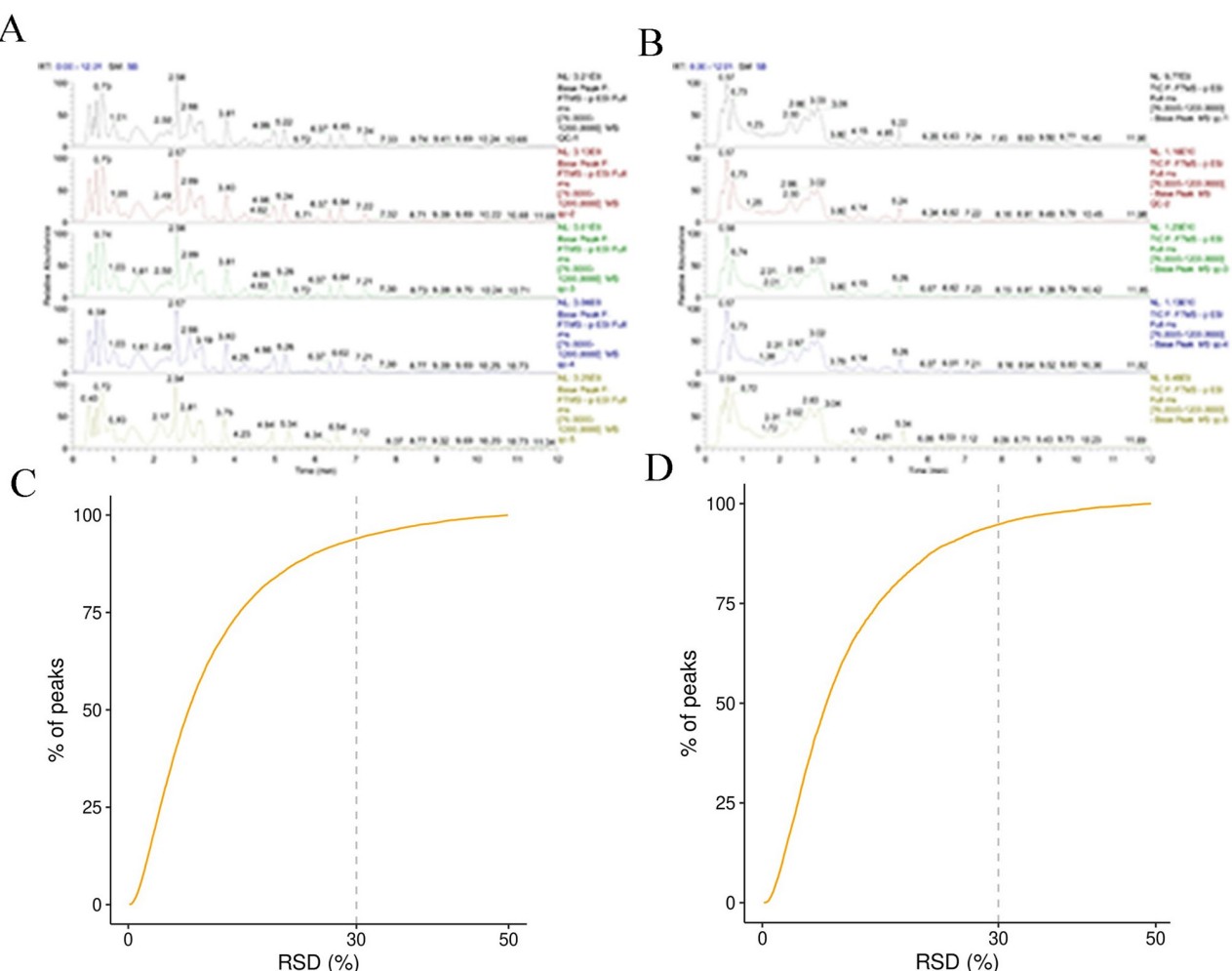

**Fig 1. Characteristics of metabolites identified by LC-MS.** (A, B) Total ion chromatogram of metabolites in positive and negative ion mode, respectively. (C, D) Quality assessment of QC samples in positive (POS) and negative (NEG) ion mode, respectively.

### Changes of metabolites in the yolk during embryonic development

In total, 636 significant metabolites were identified (VIP>1, $P<0.05$). To demonstrate the variation in metabolites between adjacent groups, the volcano plot was used (FC>1.5, $P<0.05$) (Fig 4). In the four groups that were compared, the most upregulated different metabolites (DEs) were found at E18 vs E23, while the most downregulated DEs were found at E12 vs E18. The significant fold change were observed for proser (FC = 4.600) and 7-methoxy-2-propyl-quinolin-4-ol (FC = 0.024) at E7 compared with E12, while mesoporphyrin (FC = 15.485) and 4,2'-dihydroxy-3,4',6'-trimethoxychalcone (FC = 0.045) and levorphanol (FC = 39.035) and fenazaquin (FC = 0.058) showed the maximum and minimum of FC at E12 compared with E18 and at E23 compared with E28, respectively. There were two significantly upregulated metabolites mefenamic acid (FC = 51.407) and 4-hydroxyisoleucine (FC = 40.383) and two downregulated metabolites ser-gly-ser (FC = 0.030) and methylergonovine (FC = 0.017) identified at E18 compared with E23. These findings suggested several metabolites from E18 and E23 that may be caused by the excessive metabolism during the development of goose embryo (S3 Table).

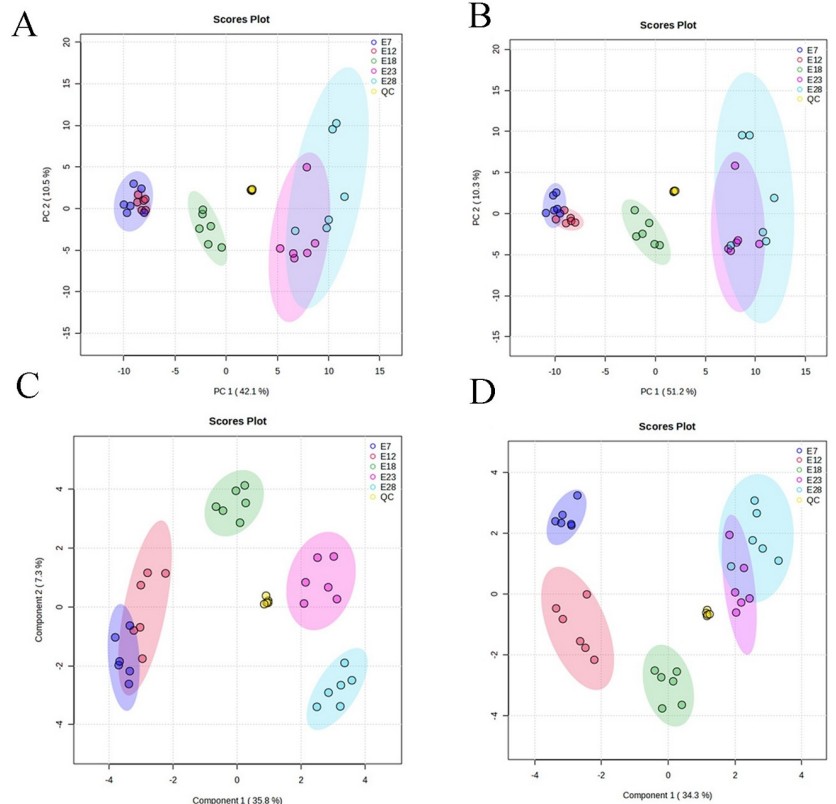

**Fig 2.** (A, B) PCA score plot. (C, D) sPLS-DA score plots in POS and NEG ion mode, respectively.

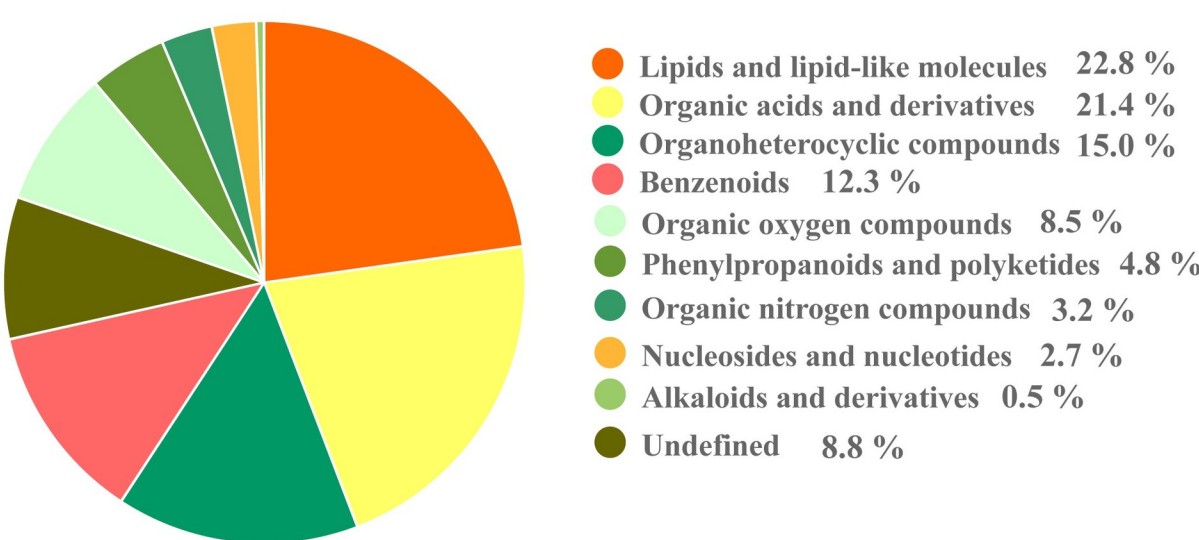

**Fig 3. Identified the major superclasses and proportion of metabolites.**

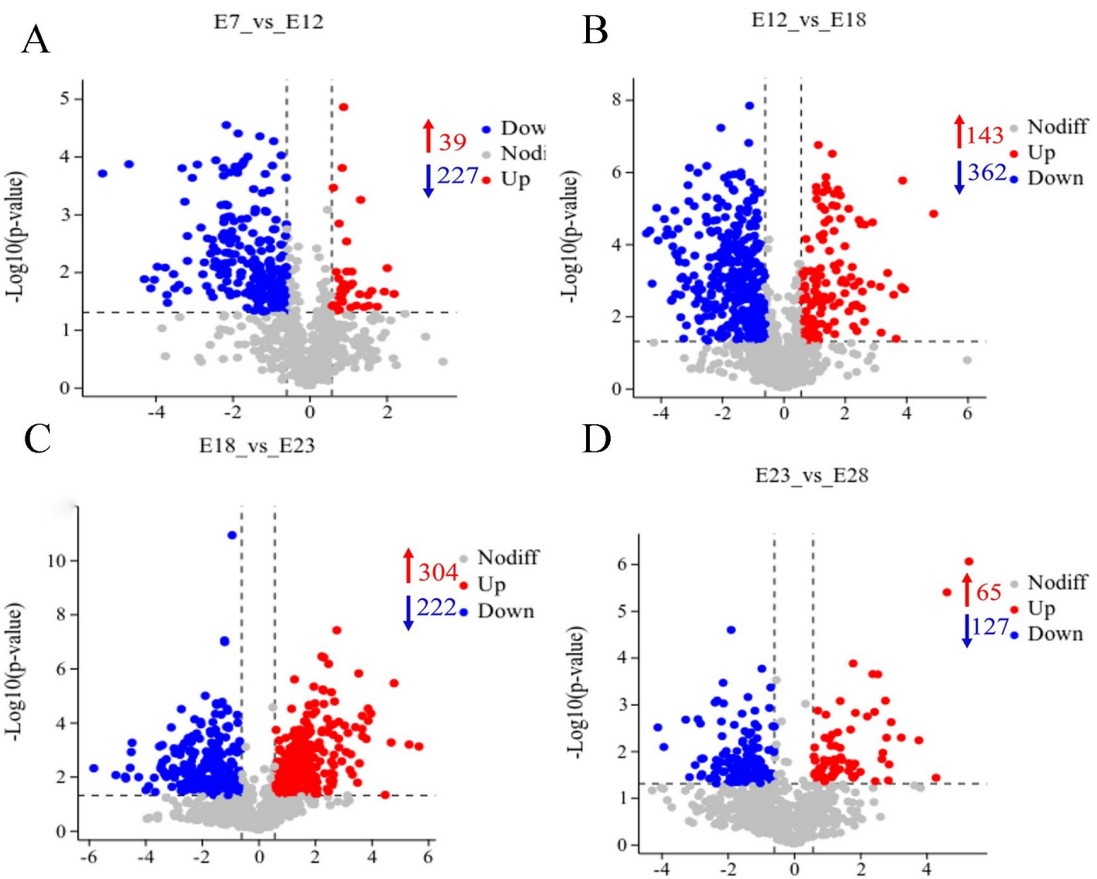

**Fig 4. Metabolites that are significantly different are displayed by red or blue dots.** While gray dots indicate no significant difference in various metabolites. Each dot corresponds to a particular metabolite.

## Metabolic events occurred during various stages of Zhijin white goose embryogenesis

According to PCA analysis, the diversity of metabolites in the egg yolk revealed a particular pattern at several periods. To elucidate the variation of metabolic pathway and biological functions during embryonic development, KEGG analysis was performed (Fig 5). From E7 to E12 and E12 to E18, identified different metabolites in the yolk were enriched primarily in biosynthesis of amino acids, valine, leucine and isoleucine, digestion and absorption of proteins, arginine biosynthesis and citrate cycle (TCA cycle). It is worth noting that both alanine, aspartate and glutamate metabolism and pyrimidine metabolism during this period. From E18 to E23, the major metabolic events were protein digestion and absorption, biosynthesis and metabolism of amino acids, aminoacyl-tRNA bio-synthesis and mineral absorption. From E23 to E28, there were fewer metabolic pathways, namely phosphotransferase system (PTS), cholesterol metabolism and pyrimidine metabolism. In general, pyrimidine metabolism was presented in the embryonic stage of development that we studied. Biosynthesis and metabolism amino acids are essential to embryonic growth and development.

## Discussion

Based on the PCA results under POS and NEG, the results that E18 was significantly separated from the other groups, E7 was similar to E12, and there is some similarity between

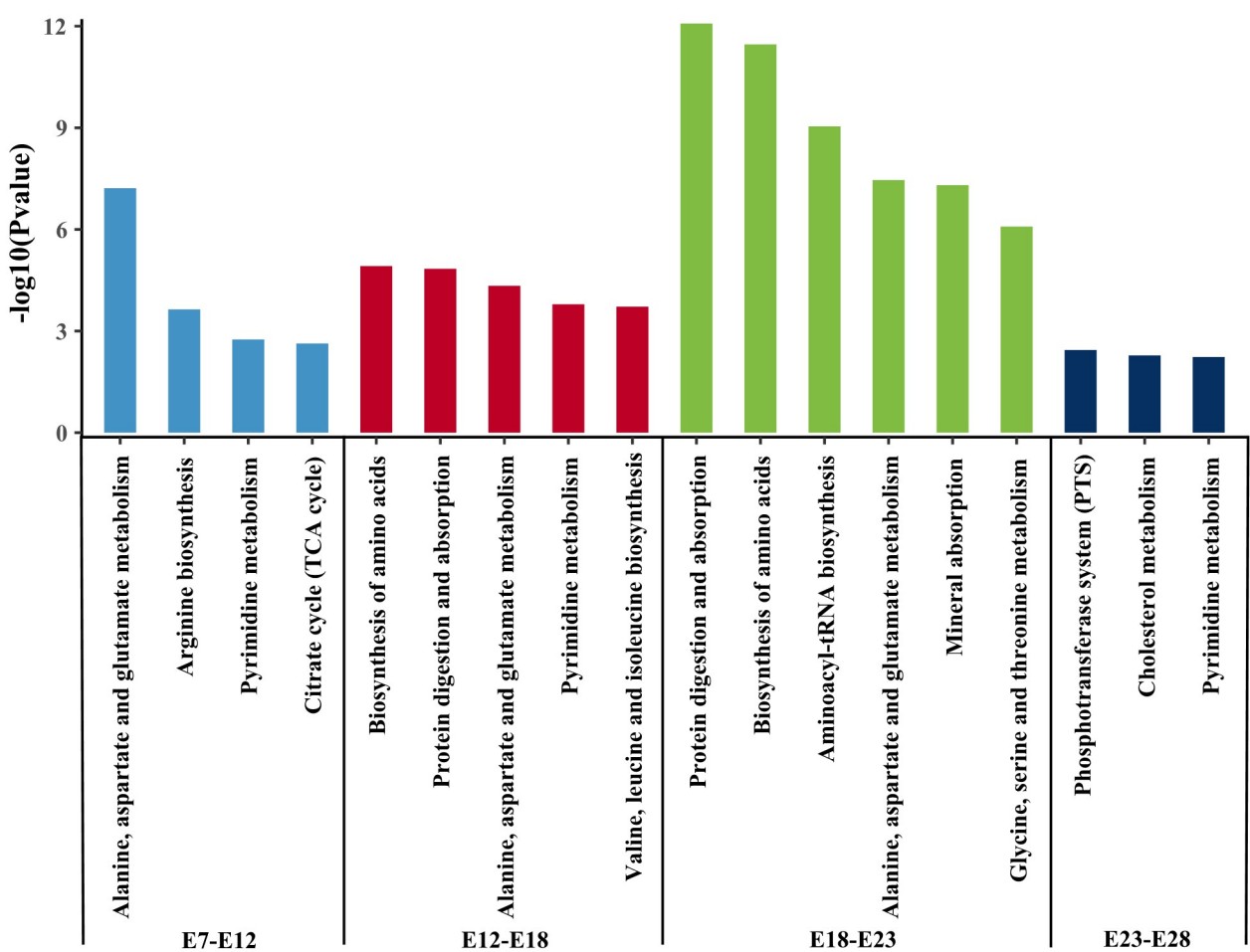

**Fig 5. The major KEGG metabolic pathways were enriched by differential metabolites between adjacent groups.** For example, E7-E12 means that when compared E7with E12 between the two groups, the differential metabolites were identified and enriched for KEGG pathway analysis according to the set parameters. Y axis indicates -log10 (*P* value). The color differences of the bar chart indicated different stages.

single samples of E23 and E28 in both modes. The results also indicated that the metabolites in egg yolk at E7 were not very different from those at E12. And the egg yolk metabolites of E23 did not change much compared to E28. Of course, the similarity between E23 and E28 may be caused by sample collection. As a whole, each group of samples at different stages of development is basically clustered together, indicating that the samples were appropriately representative. During the period of the first week of hatching, the highly vascularized yolk sac membrane (YSM) and the chorio allantoic membrane (CAM), which plays the role of respiratory organ, are not fully developed, so the nutrients in the yolk remain largely unchanged [13, 19]. After E10, YSM and CAM were completely developed and utilized yolk contents at a higher rate [20]. Studies have reported an average net lipid loss of about 200 to 300 mg per day from egg yolks prior to E13. Between E15 and E21, egg yolk net lipid loss increased significantly, and during the last 2 days of incubation, the yolk lipid was absorbed at a rate of nearly 1g per day, indicating that the metabolic rate of contents in yolk gradually accelerated during embryogenesis, that the metabolic rate of yolk contents may be gradually accelerated during embryogenesis to meet these drastic physiological changes [21]. There-fore, in our results, it is reasonable to cluster E12 and E18, E23 and E28 into different

groups according to their metabolic characteristics and the rules of embryonic development.

The changes in yolk metabolites at different stages of embryogenesis followed specific rules based on the cluster analysis results. This indicates that the dynamic metabolism of nutrients in the yolk is caused by the yolk sac membrane enclosing the yolk. To better interpret the variations of nutrients in egg yolk at different periods of embryogenesis, major metabolic pathways related to differential metabolites were analyzed. From E7 to E12, the differential metabolites identified in egg yolk were mainly enriched in the metabolism of alanine, aspartate and glutamate, arginine biosynthesis, pyrimidine metabolism and TCA cycle, indicating that protein and DNA synthesis and energy storage are mainly carried out during this stage. With the gradual maturation of the biological structure and the perfection of the function of the yolk sac, including high vascularization and high differentiation of endodermal epithelial cell (EEC) [15], the yolk sac can absorb and transport nutrients from the yolk for cell proliferation or metabolism.

In addition, creatine, which is produced from arginine, plays a crucial role in the development of the muscle and brain during the period [22, 23]. From E12 to E18, the main metabolic activities included biosynthesis and metabolism of amino acids, digestion and absorption of protein and pyrimidine metabolism. Because of the maturing of the yolk sac, it will synthesis and absorb more nutrients to promote the growth of embryos [14, 24]. Valine, leucine and isoleucine are branched chain amino acids that played significant roles in protein synthesis and brain activity of goose embryos [25]. This may be due to the fact that their metabolites are essential for the embryo's cognitive abilities, feeding and other behaviors that are required for the incubation process [26, 27]. From E18 to E23, the main metabolic activities were aminoacyl-tRNA biosynthesis and mineral absorption. As the enlarged surface area of the EEC and the formation of epithelial cell microvilli of YS, the microvilli increased the surface area of the interaction with yolk and promoted the digestion and absorption of the embryo [28, 29]. Aminoacyl-tRNA synthetases are an important family of widely distributed enzymes that help ensure accurate translation of the genetic code and play a crucial role in protein synthesis to meet the rapid synthesis of body proteins at this stage [30–32]. Minerals are important components of animal structure, such as calcium (Ca) and phosphorus (P), which are indispensable for growth and bone health [33–35]. Therefore, rapid absorption of minerals sets the stage for later embryonic movement. From E23 to E28, with the function of blood circulation and nutrient absorption of yolk sac weakened [36], lysosome and cholesterol metabolism were found as crucial metabolic pathway during the stage. At the later stage of incubation, ferroptosis is identified to be the main metabolic pathway and it is a new cell death pathway first discovered in 2012, which is different from cell death pathways such as necrosis, apoptosis and autophagy [37–39]. This is similar to the role of lysosome metabolic variation in the work. In addition, cholesterol metabolism was highlighted from E23to E28, which may indicate the formation of brain and other tissues in preparation for incubation and disease control [40–42].

## Conclusion

Egg yolk is the main nutrient source of goose embryo during hatching, but its metabolic changes during embryonic development are still not clear. Thus, we performed metabolomic analysis of the egg yolk during goose embryogenesis. The results showed that 1472 metabolites were detected, among which 264 were up-regulated and 372 were down-regulated. The principal component analysis showed that the metabolites in egg yolk were dynamically changed with embryonic development and KEGG pathway enrichment showed that the 636 differential metabolites were mainly enriched in the metabolic pathways including metabolism of

pyrimidine, cholesterol and amino acids; biosynthesis of aminoacyl-tRNA and amino acids; citrate cycle (TCA cycle); protein digestion and absorption; mineral absorption and phospho-transferase system (PTS). Amino acid metabolism and synthesis is one of the important metabolic pathways throughout embryonic development, which may indicate that amino acids played an essential role in the development of the goose embryo and yolk sac formation. Our findings may provide new insights into goose embryo nutrition and health.

## Supporting information

**S1 Table. Number of metabolites detected under each superclass.**
(XLSX)

**S2 Table. Metabolites detected in POS and NEG, respectively.**
(XLSX)

**S3 Table. The significant metabolites between adjacent groups.**
(XLSX)

## Acknowledgments

We thank everyone for their contributions to the manuscript.

## Author Contributions

**Conceptualization:** Zhonglong Zhao, Zhong Wang, Yong Zhang.

**Data curation:** Zhonglong Zhao, Zhaobi Ai, Runqian Yang.

**Formal analysis:** Zhiwei Wang.

**Funding acquisition:** Hong Yang, Yong Zhang.

**Investigation:** Zhonglong Zhao.

**Methodology:** Zhonglong Zhao, Yong Zhang.

**Supervision:** Kaibin Fu.

**Visualization:** Yong Zhang.

**Writing – original draft:** Zhonglong Zhao.

**Writing – review & editing:** Zhiwei Wang, Tiansong Wang, Yong Zhang.

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
