## [Decision Letter · Decision Letter 0]

26 Dec 2023

PONE-D-23-38791Metabolomics analysis of the yolk of Zhijin white goose during the embryogenesis based on LC-MS/MSPLOS ONE

Dear Dr. Zhang,

Thank you for submitting your manuscript to PLOS ONE. After careful consideration, we feel that it has merit but does not fully meet PLOS ONE’s publication criteria as it currently stands. Therefore, we invite you to submit a revised version of the manuscript that addresses the points raised during the review process.

The submitted manuscript titled "Metabolomics analysis of the yolk of Zhijin white goose during the embryogenesis based on LC-MS/MS"  can be considered for publication in "Plos One" journal with minor revision. Authors require some minor modification in submitted manuscript as suggested by the learned reviewers.  Reviewer's comments are as follows:

Reviewer #1: Finding of the reviewed manuscript will give new source of information to the scientific community. It is relevant work and also good effort from the author's side. So the manuscript may be accepted for possible publication in the Plos One after considering all the formalities and rules and regulations of the Plos One publication policy as well overall publication guidelines.

Reviewer #2: In this manuscript, authors studied the “Metabolomics analysis of the yolk of Zhijin white goose during the embryogenesis based on LC-MS/MS”. It is a good piece of research that will work for understanding about embryonic development. Authors have studied about the identification of metabolites and metabolite changes in egg yolk that provides nutrient requirement in goose embryonic development. In this review, authors have summarized that may provide new ideas for improving prehatch embryonic health and nutrition.

Authors should ensure that they have followed the Instructions to Authors of this journal. Overall, the manuscript in its current form is not acceptable for publication in the esteemed “Plos One” journal. Authors require minor revisions in the manuscript and can submit the revised manuscript. Some important comments listed as follows:

1. On Page 2, Line 26, please correct E (7) and define the numbers.

2. In abstract, authors have written that yolk metabolites were approximately similar at E7 and E12, E23 and E28 but what about E18.

3. Abbreviations used first time in the text should be explanatory (Page 3, Line 56).

4. The format of heading should be uniform throughout manuscript.

5. Please cite the references form where extraction procedure and other methodologies were adopted.

We look forward to receiving your revised manuscript.

Kind regards,

Pankaj Singh, Ph.D.

Academic Editor

PLOS ONE

Journal Requirements:

"The Science and Technology Project of Guizhou Province, China (No. QKHZC[2021]YB156 and No. QKHZC[2020]1Y047)."

"This work was supported by two grants received from the Science and Technology Project of Guizhou Province, China (No. QKHZC[2021]YB156 and No.QKHZC[2020]1Y047)."

"The Science and Technology Project of Guizhou Province, China (No. QKHZC[2021]YB156 and No. QKHZC[2020]1Y047)."

5. Please provide a complete Data Availability Statement in the submission form, ensuring you include all necessary access information or a reason for why you are unable to make your data freely accessible. If your research concerns only data provided within your submission, please write "All data are in the manuscript and/or supporting information files" as your Data Availability Statement.

Reviewers' comments:

Reviewer's Responses to Questions

**Comments to the Author**

1. Is the manuscript technically sound, and do the data support the conclusions?

Reviewer #1: Yes

Reviewer #2: Yes

2. Has the statistical analysis been performed appropriately and rigorously? 

Reviewer #1: Yes

Reviewer #2: Yes

3. Have the authors made all data underlying the findings in their manuscript fully available?

Reviewer #1: Yes

Reviewer #2: Yes

4. Is the manuscript presented in an intelligible fashion and written in standard English?

Reviewer #1: Yes

Reviewer #2: Yes

5. Review Comments to the Author

Reviewer #1: Finding of the reviewed manuscript will give new source of information to the scientific community. It is relevant work and also good effort from the author's side. So the manuscript may be accepted for possible publication in the Plos One after considering all the formalities and rules and regulations of the Plos One publication policy as well overall publication guidelines.

Reviewer #2: In this manuscript, authors studied the “Metabolomics analysis of the yolk of Zhijin white goose during the embryogenesis based on LC-MS/MS”. It is a good piece of research that will work for understanding about embryonic development. Authors have studied about the identification of metabolites and metabolite changes in egg yolk that provides nutrient requirement in goose embryonic development. In this review, authors have summarized that may provide new ideas for improving prehatch embryonic health and nutrition.

Authors should ensure that they have followed the Instructions to Authors of this journal. Overall, the manuscript in its current form is not acceptable for publication in the esteemed “Plos One” journal. Authors require minor revisions in the manuscript and can submit the revised manuscript. Some important comments listed as follows:

1. On Page 2, Line 26, please correct E (7) and define the numbers.

2. In abstract, authors have written that yolk metabolites were approximately similar at E7 and E12, E23 and E28 but what about E18.

3. Abbreviations used first time in the text should be explanatory (Page 3, Line 56).

4. The format of heading should be uniform throughout manuscript.

5. Please cite the references form where extraction procedure and other methodologies were adopted.

6. PLOS authors have the option to publish the peer review history of their article (what does this mean?). If published, this will include your full peer review and any attached files.

Reviewer #1: No

Reviewer #2: No

---

## [Author Response · Author response to Decision Letter 0]

2 Jan 2024

Dear editors and reviewers:

Thank you very much for your careful review and constructive suggestions with regard to our manuscript “Metabolomics analysis of the yolk of Zhijin white goose during the embryogenesis based on LC-MS/MS (PONE-D-23-38791R1)”. We have carefully evaluated the Editors/Reviewers’ critical comments and thoughtful suggestions, responded to these suggestions point-by-point, and revised the manuscript accordingly. The revised parts in the manuscript have been marked in red. We appreciate for Editors/Reviewers’ warm work earnestly, and hope that the corrections will meet with approval. Please feel free to contact us with any questions and we are looking forward to your consideration. The main corrections in the manuscript and the responses to the Editors/Reviewers’ comments are as follows.

1. On Page 2, Line 26, please correct E (7) and define the numbers.

Response: Thank you very much, we have corrected E (7) and defined the numbers.

2. In abstract, authors have written that yolk metabolites were approximately similar at E7 and E12, E23 and E28 but what about E18.

Response: According to the advice, we have made appropriate changes to this sentence and added a note about E18.

3. Abbreviations used first time in the text should be explanatory (Page 3, Line 56).

Response: According to the advice, we have explained the acronyms EPA and DHA.

4. The format of heading should be uniform throughout manuscript.

Response: Based on the PLOS ONE style template, we have changed the level 1 headings to 18pt font and bold type, and the level 2 headings to 16-point font and bold type throughout manuscript.

5. Please cite the references form where extraction procedure and other methodologies were adopted.

 Response: Thank you very much, we have cited the corresponding references in the section on metabolite extraction and data processing.

We have tried our best to improve the manuscript and made some changes in the manuscript. We appreciate for Editors/Reviewers’ warm work earnestly, and hope that the correction will meet with approval. Once again, thank you very much for your comments and suggestions.

Yours sincerely

---

## [Decision Letter · Decision Letter 1]

5 Jan 2024

Metabolomics analysis of the yolk of Zhijin white goose during the embryogenesis based on LC-MS/MS

PONE-D-23-38791R1

Dear Dr. Zhang,

We’re pleased to inform you that your manuscript has been judged scientifically suitable for publication and will be formally accepted for publication once it meets all outstanding technical requirements.

Kind regards,

Pankaj Singh, Ph.D.

Academic Editor

PLOS ONE

Additional Editor Comments (optional):

Reviewers' comments:

Reviewer's Responses to Questions

**Comments to the Author**

1. If the authors have adequately addressed your comments raised in a previous round of review and you feel that this manuscript is now acceptable for publication, you may indicate that here to bypass the “Comments to the Author” section, enter your conflict of interest statement in the “Confidential to Editor” section, and submit your "Accept" recommendation.

Reviewer #2: All comments have been addressed

2. Is the manuscript technically sound, and do the data support the conclusions?

Reviewer #2: Yes

3. Has the statistical analysis been performed appropriately and rigorously? 

Reviewer #2: Yes

4. Have the authors made all data underlying the findings in their manuscript fully available?

Reviewer #2: Yes

5. Is the manuscript presented in an intelligible fashion and written in standard English?

Reviewer #2: Yes

6. Review Comments to the Author

Reviewer #2: The revised manuscript entitled “Metabolomics analysis of the yolk of Zhijin white goose during the embryogenesis based on LC-MS/MS” submitted for publication can be accepted for publication in esteemed "Plos One" journal. Authors have addressed each queries very well raised by reviewer and have critically modify the manuscript as per requirement.

7. PLOS authors have the option to publish the peer review history of their article (what does this mean?). If published, this will include your full peer review and any attached files.

Reviewer #2: No

---

## [Editor Report · Acceptance letter]

18 Jan 2024

PONE-D-23-38791R1 

PLOS ONE

Dear Dr. Zhang, 

I'm pleased to inform you that your manuscript has been deemed suitable for publication in PLOS ONE. Congratulations! Your manuscript is now being handed over to our production team.

Kind regards, 

on behalf of

Dr. Pankaj Singh 

Academic Editor

PLOS ONE